# The Effect of External Cephalic Version on Fetal Circulation: A Prospective Cohort Study

**DOI:** 10.3390/children10020354

**Published:** 2023-02-10

**Authors:** Offra Engel, Shmuel Arnon, Gil Shechter Maor, Hanoch Schreiber, Ettie Piura, Ofer Markovitch

**Affiliations:** 1Obstetrical & Gynecological Ultrasound Unit, Department of Obstetrics and Gynecology, Meir Medical Center, Kfar Saba 4428164, Israel; 2Sackler Faculty of Medicine, Tel Aviv University, Tel Aviv 6997801, Israel; 3Department of Neonatology, Meir Medical Center, 4428163 Kfar Saba, Israel; 4High Risk Pregnancy Unit, Department of Obstetrics and Gynecology, Meir Medical Center, Kfar Saba 4428164, Israel

**Keywords:** external cephalic version, umbilical artery, doppler, velocimetry, placenta, fetal circulation, acute stress

## Abstract

External cephalic version (ECV) is a cost-effective and safe treatment option for breech presentation at term. Following ECV, fetal well-being is assessed via a non-stress test (NST). An alternative option to identify signs of fetal compromise is via the Doppler indices of the umbilical artery (UA), middle cerebral artery (MCA) and ductus venosus (DV). Inclusion criteria were an uncomplicated pregnancy with breech presentation at term. Doppler velocimetry of the UA, MCA and DV were performed up to 1 h before and up to 2 h after ECV. The study included 56 patients who underwent elective ECV with a success rate of 75%. After ECV, the UA S/D ratio, UA pulsatility index (PI) and UA resistance index (RI) were increased compared to before the ECV (*p* = 0.021, *p* = 0.042, and *p* = 0.022, respectively). There were no differences in the Doppler MCA and DV before or after ECV. All patients were discharged after the procedure. ECV is associated with changes in the UA Doppler indices that might reflect interference in placental perfusion. These changes are probably short-term and have no detrimental effects on the outcomes of uncomplicated pregnancies. ECV is safe; yet it is a stimulus or stress that can affect placental circulation. Therefore, careful case selection for ECV is important.

## 1. Introduction

Breech presentation is the most common non-vertex presentation and one of the main indications for Cesarian section (CS), as vaginal delivery in this presentation is correlated with higher fetal morbidity and mortality [1]. However, Cesarean section is one of the most significant contributing factors to postpartum maternal morbidity in developed countries, and is known to cause significant complications, which can sometimes result in permanent health impairments. External cephalic version (ECV) can provide a safe, economical, and relatively simple solution to avoid surgical delivery [2]. As such, it is a standard of care recommended by the American College of Obstetricians and Gynecologists and the Royal College of Obstetricians and Gynecologists as first line management of every term breech presentation pregnancy without contraindications [3].

ECV is performed by manipulating the abdomen in a circular motion, rotating the fetal head to vertex. The procedure is successful in 58% of attempts, reduces the risk for CS by two-thirds, and enables 80% of these patients to eventually deliver vaginally [2]. In a 2015 Cochrane systematic review, compared to a caesarian section, ECV did not result in significant differences in adverse perinatal outcomes, including in NICU hospitalizations, neonatal mortality and Apgar scores [1]. Nevertheless, ECV encompasses its own risks, including temporary uterine activity (5–9%) [4], temporary bradycardia of the baseline fetal heart rate (5.7%) [5,6], fetal heart rate decelerations (0.2%) [5], placental abruption (0.12%) [4], vaginal bleeding (0.4%) [4], and maternal and fetal bleeding (2.4%) [7]. Based on the low rate of complications, a Cochrane review concluded that ECV is considered a safe procedure and can be recommended [1]. Therefore, it is routinely performed by obstetricians in many delivery rooms worldwide.

Several prognostic parameters are associated with the success of ECV, including a higher parity [8], amniotic fluid index over 7 cm, estimated fetal weight [8], non-anterior placental location [9], type of breech presentation [10], and maternal BMI [11].

After ECV is performed, fetal wellbeing is assessed via prolonged fetal heart rate monitoring (NST) and an ultrasound biophysical profile. Transient abnormal cardiotocography has been described after ECV; however, it does not accurately predict fetal mortality and seldom leads to the need for cesarean section [8,9,10]. Other subclinical side effects of ECV, such as an increase in cell-free fetal DNA content in the maternal circulation [12] and decreased pulsatility index (PI) in the middle cerebral artery (MCA) [13] have been described.

Doppler ultrasound is another important tool for evaluating fetal well-being. Doppler ultrasound technology is based on sound waves detecting blood flow in vessels (e.g., red blood cells) [10] and is often described as PI in the MCA, the umbilical artery (UA), and the ductus venosus (DV) [11,14].

Its main applications are in high-risk pregnancies, to detect fetal anemia and manage fetuses with intrauterine growth restriction (IUGR) [15]. It also has a prognostic role in twin-to-twin transfusion syndrome undergoing in utero intervention [15]. Since placenta-based IUGR is predominantly a vascular disorder, its effects can be documented via a Doppler ultrasound examination of several vessels: the maternal uterine arteries and the fetal UA for the placenta; the MCA for preferential brain perfusion; and the precordial veins for the cardiac effects of placental dysfunction [16]. As IUGR worsens, Doppler abnormalities in these vascular territories also deteriorate, suggesting a sequential pattern of disease progression [17,18]. However, the characteristics of cardiovascular manifestations in IUGR are determined by the gestational age at onset and the severity of the placental disease [17].

In pregnancies with early IUGR, the waveforms and the angle-independent indices become abnormal, and the changes allow the physician to track the physiological compensatory fetal state and decide how to best manage the pregnancy [15,17]. These changes reflect increased placental vascular resistance [15].

Using a Doppler examination as a screening test for fetal wellbeing is still controversial due to the possibility that integrating this examination may lead to unnecessary interventions and adverse obstetrical outcomes [19].

It has been demonstrated that in acute hypoxia, the proportion of placental blood passing through the DV increases, which reduces the placental fraction of the cardiac output [18] and promotes adenosine release. This depresses fetal cerebral oxygen consumption and vasodilatation [16].

Information regarding Doppler indices following ECV is scarce. Lau et al. [13] used Doppler to examine changes in fetal circulation after ECV in 136 women who underwent the procedure at ≥36 weeks of pregnancy. The study found no significant difference in the PI of the UA before and after ECV. It found a significant decrease in the PI of the MCA after ECV among multiparous women, and in cases of posterior placentation or difficult ECV. In addition, the amount of pressure exerted in the ECV was related to changes in the fetal circulation [20]. The greater the force applied during external cephalic version, the greater the decrease in the pulsatility indices of the MCA and UA, indicating an increase in blood flow through these arteries. Since these studies were published, the Doppler technique has advanced technologically, and has become more sensitive and accessible. As ECV has become more common [2], the question of the role of Doppler in fetal assessment after the procedure remains. The primary objective of this study was to investigate the effect of ECV on fetal circulation by measuring Doppler indices in the MCA, UA, and DV. The secondary objective was to assess the effect of routine vs. difficult ECV on clinical and Doppler parameters.

## 2. Materials and Methods

This prospective observational study included 56 women who underwent ECV, in a tertiary university teaching hospital, over a 6-month period. The cohort consisted of women who were referred for ECV after being diagnosed with a fetus in a non-vertex position; inclusion criteria were gestational age of at least 36 completed weeks, an uncomplicated pregnancy, normal fetal cardiotocography prior to the ECV, and normal amniotic fluid index on ultrasound examination. Exclusion criteria included patients who did not wish to be included or asked to stop their participation without completing the post-ECV examination, or any contraindication for ECV (oligohydramnios, premature rupture of membranes, etc.).

The procedure took place in a designated room in the labor and delivery complex, only in the morning and early afternoon hours, by an assigned team. After admission, reassessment of the normal biophysical profile and verification of the non-vertex position were performed clinically and via ultrasound. Data regarding socioeconomic and clinical characteristics, gestational and previous delivery course, Doppler sonographic measurements of the fetal UA, MCA, and DV circulation were recorded. Fetal cardiotocography were placed for at least 20 min; if fetal well-being was not established after 20 min, monitoring was extended until satisfactory. A senior obstetrician attempted the ECV procedure. During the process, the fetal cardiotocography was disconnected but not removed and was replaced after the attempt was completed.

In cases where the first attempt failed and the clinical impression of the performing physician was that tocolysis might assist the procedure, 0.025 mg ventolin was given subcutaneously to relax the uterine smooth muscle, and the procedure was defined as complicated. The procedure was attempted again 20 min later. After successful ECV, a second NST was performed for at least 1 h to establish fetal well-being and no significant uterine activity. If both conditions were met, the patient was discharged. Otherwise, the fetal examination was extended, or the patient was hospitalized for longer surveillance and repeated monitoring. In the rare cases where severe fetal distress was established, an emergency CS was performed.

The ultrasound examination, including the Doppler studies, were performed up to 1 h before and up to 2 h after the ECV. All ultrasound studies were performed by physicians who were certified sonographers. A GE Voluson P6 ultrasonography machine with a 4C convex transabdominal probe, with 2–5.5 MHz band width was used for all examinations. All Doppler studies were performed in the absence of gross fetal body or breathing movements. A pulsed Doppler gate was optimized according to the diameter of the vessel examined until a regular circulation wave pattern appeared (0.5–1.5 mm in the UA measurement, 1.5–2.5 mm in MCA, and 0.5–1.0 mm for DV, with a sample volume of 1–3 mm for all). After the examiner verified the quality and accuracy of the waves, they were measured automatically by the machine. All measurements were printed for documentation. Delivery and neonatal characteristics were documented postpartum.

The maternal, ECV, and delivery characteristics are depicted in Table 1.

### Data Analysis

All patients who met the inclusion criteria were included in the statistical analysis. The nonparametric paired Wilcoxon signed-rank test was used to compare the pre- and post-ECV Doppler results. This test was used to accurately compare the median values of continuous non-normally distributed data generated via Doppler examination and considering the cohort size, since it does not require any assumptions of normality or equal variances. A *p*-value of <0.05 was considered significant.

## 3. Results

The study included 56 patients who underwent elective ECV. The overall success rate was 75%. There were no significant differences in the socioeconomic variables and pregnancy complications between women who experienced successful or failed ECV. In the group who underwent successful ECV but ultimately delivered via CS were two patients who presented with recurrent non-vertex presentation in the presence of latent or active phase labor, and eight with nonreassuring fetal heart rates requiring urgent delivery. No patient had an indication for urgent CS immediately after ECV.

All Doppler sonographic measurements of the circulation of the fetal UA, MCA, and DV were documented. The measurements are detailed in Table 2 (regardless of the ECV outcome, *n* = 56).

Doppler studies of the MCA, UA, and DV were classified according to the level of difficulty of the procedure, either as routine, complicated, or failed (Table 3).

In the MCA studies, there was no significant difference between the pre- and post-ECV Doppler results when comparing the entire cohort as one group. When stratifying the procedure according to the level of difficulty of ECV, there was a significant decrease in the PI (*p* = 0.03) and RI (*p* = 0.3) of the MCA after a complicated ECV (Table 3).

In the UA studies, ECV was associated with significant disturbance of placental resistance to blood flow: UA S/D ratio (*p* = 0.021), UA PI (*p* = 0.042), and UA resistance index (RI) (*p* = 0.022) (Table 2). When analyzing the data according to the difficulty of the procedure, there was no significant difference in any of the measurements mentioned.

There was no significant difference between the pre- and post-ECV Doppler results in the studies of the DV, regardless of whether the comparison was in the entire group or stratified for parity, outcome of ECV, or its level of difficulty (Table 3).

As expected, there was a significant association between the level of difficulty of the procedure and the duration of the ECV (14.8 ± 7.7 min, *p* = 0.006), number of attempts (1.9 times, *p* = 0.001), number of physicians involved (2.4, *p* = 0.001), and CS delivery (0.007). There was also a significant correlation between the need for manual lysis of the placenta at birth (*p* = 0.04).

There was no significant difference between the level of difficulty and the pregnancy and delivery characteristics of the maternal age (*p* = 0.27), maternal height (*p* = 0.49), BMI at presentation (*p* = 0.7), weight gained during the pregnancy (*p* = 0.73), gestational age (*p* = 0.73), amniotic fluid index (*p* = 0.56), estimated fetal weight (*p* = 0.99), neonatal blood pH at birth (7.2, *p* = 0.31), delivery weight (*p* = 0.69), 5-min Apgar score (*p* = 0.27), meconium stained amniotic fluid (*p* = 0.62), placental abruption (*p* = 1), post-partum hemorrhage (*p* = 1), true umbilical cord knot (*p* = 0.28), or nuchal cord at birth (*p* = 0.31).

Among the 56 cases, there was one case of placental abruption (1.7%). There was no fetal mortality or morbidity attributable to the ECV.

## 4. Discussion

This study describes the effect of ECV on fetal circulation in the MCA, UA, and DV, measured via Doppler ultrasound.

Our findings revealed significant decreases in the RI, PI, and S/D ratio in the UA following ECV. These changes represent the placental resistance and suggest that the ECV mainly affected the placental arterial vessels [21]. We speculate that the placenta functions as a vascular organ that reacts to direct external pressure and that the arterial system reflects these changes. In previous studies, the effects of ECV on fetal arterial circulation was found to be inconsistent, ranging from no disturbance of placental resistance to blood flow mainly when the placenta was located posteriorly and to increases in the umbilical blood flow when it was located laterally [13,20]. These studies argued that these differential effects of placentation on fetal blood flow suggest that the vascular effects are more likely to be secondary to the direct exertion of force on the placenta instead of a general reaction to ECV, in which case the fetal heart rate would also be expected to increase. In addition, they reported that the amount of pressure exerted also played a role in the effect on fetal circulation. An in vitro study proposed that placentas have a non-linear, perfusion pressure-dependent resistance, and that increased external pressures increased the placental resistance at lower perfusion pressures [22].

Nevertheless, several previous studies demonstrated that ECV is a safe procedure [2,5,23]. This was confirmed in our study as well: there was no fetal mortality or morbidity attributable to the ECV. Hence, we can safely assume that these changes are short-term and do not have a detrimental effect on fetal morbidity and mortality.

Furthermore, the venous system, represented in this study by the Doppler studies in the DV, was not affected by the ECV, regardless of whether the comparison was in the entire group or stratified according to the difficulty of the procedure.

To the best of our knowledge, this is the first report to assess venous fetal circulation during ECV. The waveform in the DV has two peaks representing chronological hemodynamic changes in pressure gradients between the umbilical vein and the right atrium [24]. The effect of disturbances in arterial circulation when stress is encountered is also demonstrated in cases of chronic stress due to placental insufficiency, such as fetal growth restriction or preeclampsia [25,26]. Early-onset fetal growth restriction before 34 weeks of gestation has a characteristic sequence of responses to placental dysfunction that evolves from the arterial circulation to the venous system and finally to biophysical abnormalities [25,26]. In these cases, the DV has an important role in circulatory adaptation to hypoxia. Under hypoxic conditions, increased blood shunting through the DV has been demonstrated [27]. In practice, DV PI is thought to be one of the most useful parameters with which to decide to deliver a decompensated fetus at the appropriate time [28] and that delaying delivery until DV changes occur will result in the most favorable neurodevelopmental outcome for the offspring [29,30].

However, in normal pregnancies, there is no conclusive evidence that routine Doppler ultrasound exams affect neonatal outcomes. There is some evidence that UA Doppler velocimetry may reduce the risk of potentially preventable perinatal deaths, but these findings are based on a few trials and subgroup analyses [31]. Also there is evidence that the cerebroplacental ratio can be used as a predictor for adverse perinatal outcomes in AGA term pregnancies [32,33] and in late IUGR [34]. The cerebroplacental ratio is calculated by dividing the PI in the MCA by the PI in the UA, as this ratio reflects both the preferential supply to the brain of the fetus and the increase in placental resistance [35].

The literature discussing the effect of acute stress on the fetoplacental circulation is sparse. Nevertheless, two prevalent risk factors have been investigated. Maternal smoking was correlated with acute circulatory changes, manifested by changes in the UA flow and diameter, and chronic or acute fetal anemia, which is seen as changes in the flow velocity and diameter of MCA [36,37]. The current study presents another potential cause for acute circulatory changes in the fetoplacental circulation. In the case of ECV, these alternations were reversible.

Cases defined as difficult ECVs were correlated with significant reductions in the resistance measurements of RI and PI in the MCA (both *p* = 0.03). These vascular changes in cerebral arterial blood flow were unlikely to be related to the direct effects of the ventolin used for uterine relaxation before the ECV, since it has been shown that there is no effect on the systolic/diastolic ratios of the uterine arcuate, umbilical, or middle cerebral arteries during the first 2 h after administration [38,39]. This finding is also supported by two studies that investigated the effect of ECV on fetal circulation. Lau et al. presumed this was the response to the pressure on the fetus and occurred only when the procedure was difficult, suggesting that the stimulation is related to direct pressure on the fetus. Several studies established that MCA blood flow was reduced when pressure was applied to the fetal head [40,41,42]. Leung et al. suggested that there may be a significant decrease in cerebral blood flow during ECV, and to some extent, transient cerebral ischemia may result. Therefore, after the pressure is released, there may be a compensatory rebound of hyperemia, as reflected by an increase in the MCA PI. We would like to suggest that this effect is due to an adaptive reaction to counter fetal hypoxia, known as the brain sparing effect, usually demonstrated in fetuses with growth restriction [43]. Brain sparing manifests in chronic fetal hypoxia and is mediated by changes in resistance of the arterial vessels in the brain, as highly oxygenated blood preferentially flows through the DV towards the cerebral circulation, bypassing the liver and the lungs [44]. The brain sparing mechanism helps to ensure adequate blood supply to the developing fetal brain, even during times of reduced blood flow to the uterus [45], and depends on the fetal cardiovascular system [43]. In late gestation, the fetal cardiovascular system adopts strategies to decrease oxygen consumption and redistribute the cardiac output away from peripheral vascular beds towards essential circulations [43].

Low MCA PI correlates with fetal outcomes, not only in early severe cases accompanied by an abnormal UA PI, but also in late and term IUGR in which the UA PI might be normal [46].

When examining the effect of acute stress on the cerebral circulation, such as during uterine contractions, the MCA PI was found to decease significantly due to the vasodilatation involved in the brain-sparing effect, but was more prominent in OCT-positive cases [47]. This vasodilation in the fetal cerebral circulation occurs as result of an exclusive reaction of the carotid chemoreflex [48,49], the acute hypoxia that triggers local increases in adenosine and, to a lesser extent, nitric oxide (NO) and prostanoids [50,51].

This study had several advantages. It was prospective, all Doppler studies were performed in one of two designated beds and with cardiotocogram monitors, using the same ultrasound machine, and by only three physicians, who are certified sonographers. The ECV was performed by a small team of certified obstetricians with at least 5 years’ experience in performing ECV.

The present study also had some limitations, including the relatively short interval between the ECV and the doppler examination, which did not allow us to determine how long the effect of the ECV on fetal circulation persisted. Another limitation was the relatively small cohort.

## 5. Conclusions

ECV is associated with changes in UA Doppler indices that might reflect interference in placental perfusion. The changes in the MCA Doppler might reflect an adaptive reaction to counter fetal stress and acute hypoxia. Both changes are probably short-term and have no detrimental effect on the outcomes of healthy pregnancies.

ECV is a safe procedure, yet it affects the placental circulation. Hence, careful case selection for ECV is very important.

## Figures and Tables

**Table 1 children-10-00354-t001:** Maternal, external cephalic version (ECV), and delivery characteristics (*n* = 56).

Pregnancy Course	
Age (y), mean ± SD	31.9 ± 5.1
Height (m), mean ± SD	1.62 ± 0.07
Nulliparous, *n* (%)	22 (39.3)
BMI at presentation, mean ± SD	26 ± 6.01
Weight gained (kg), mean ± SD	11.50 ± 4.18
Gestational age (weeks), mean ± SD	37 + 6 ± 0.6
Amniotic fluid index (cm), mean ± SD	12.5 ± 5.4
Estimated fetal weight (g), mean ± SD	2954.3 ± 321.9
Chronic (pregestational) disease, *n* (%)	5 (8.9)
Gestational diabetes mellitus, *n* (%)	9 (16.8)
COVID-19 vaccinated, *n* (%)	43 (84.3)
COVID-19 recovered, *n* (%)	9 (16.7)
Anterior placentation, *n* (%)	28 (51.9)
Frank breech presentation, *n* (%)	34 (66.7)
Male sex, *n* (%)	32 (58.8)
ECV characteristics	
Duration (min)	14.76 ± 7.66
Medication for muscle relaxation, *n* (%)	32 (57)
Duration (min)	14.8 ± 7.7
Success rate, *n* (%)	42 (75)
Delivery characteristics	
Cesarian delivery, *n* (%)	24 (39.3)
Meconium-stained amniotic fluid, *n* (%)	8.2 (15.6)
Placental abruption, *n* (%)	1 (1.7)
Post-partum hemorrhage, *n* (%)	3 (5.3)
Revision/manual lysis, *n* (%)	6 (10.7)
True knot in umbilical cord, *n* (%)	3 (5.4)
Nuchal cord	
Total, *n* (%)	45 (83.3)
1 loop	14
2 loops	28
4 loops	3
Apgar score, *n* (%)	
1 min < 7	1 (1.8)
5 min < 7	1 (1.9)
Neonatal pH 7 ± 0.6	0
Birth weight (g)	3223.8 ± 463.6

ECV, external cephalic version.

**Table 2 children-10-00354-t002:** Doppler measurements before and after external cephalic version (*n* = 56).

Vessel Examined	Measurement Evaluated	Before (Mean ± SD)	After (Mean ± SD)	*p*-Value
Umbilical artery	Peak systolic velocity	39.65 ± 10.6	42.18 ± 10.8	0.289
	S/D ratio	2.4 ± 0.5	2.5 ± 0.6	0.021
	Pulsatility index	0.8 ± 0.2	0.9 ± 0.2	0.042
	Resistance index	0.6 ± 0.1	0.6 ± 0.2	0.022
Middle cerebral artery	Peak systolic velocity	43.74 ± 11.79	45.12 ± 16.1	0.451
	S/D Ratio	0.4 ± 1.9	6.8 ± 0.025	0.025
	Pulsatility index	1.4 ± 0.4	1.5 ± 0.4	0.070
	Resistance index	0.7 ± 0.1	0.8 ± 0.1	0.089
Ductus venosus	S-wave max velocity	40.7 ± 13.1	30.6 ± 15	0.230
	S/A ratio	3.2 ± 1.8	4.1 ± 6	0.975
	Pulsatility index	0.7 ± 0.1	0.7 ± 0.1	0.949
	Peak velocity index for veins	1.3 ± 2.5	3.3 ±3.2	0.623

**Table 3 children-10-00354-t003:** Differences in Doppler study indices of the middle cerebral artery, umbilical artery, and ductus venosus (DV) before and after external cephalic version (ECV), according to difficulty of the procedure.

Vessel Examined	ECV Level of Difficulty			Total (*n* = 54)	*p* Value
	Routine (*n* = 17)	Complicated (*n* = 23)	Failed (*n* = 14)		
Middle cerebral artery					
Peak systolic velocity	−0.8 ± 15.69	−0.67 ± 22.34	−4.48 ± 14.94	−1.74 ± 18.46	0.79
End diastolic velocity	−0.37 ± 4.92	1.16 ± 5.17	2.68 ± 5.8	1.1 ± 5.3	0.258
S/D ratio	−0.12 ± 2.15	1.16 ± 5.17	2.68 ± 5.8	1.1 ± 5.3	0.258
Pulsatility index	0.06 ± 0.53	−0.06 ± 0.46	−0.46 ± 0.66	−0.13 ± 0.57	0.03
Resistance index	−0.01 ± 0.01	0.01 ± 0.16	−0.11 ± 0.14	−0.02 ± 0.15	0.03
Mid-diastolic velocity	−0.8 ± 5.41	0.2 ± 5.61	0.80 ± 5.03	0.05 ± 5.34	0.48
Time-averaged maximum velocity	4.14 ± 8.63	−3.37 ± 1.63	−3.36 ± 10.17	−1.48 ± 9.46	0.47
Heart rate	3.75 ± 8.43	2.5 ± 2.71	2.71 ± 12.59	−2.96 ± 13.28	0.8
Umbilical Artery					
Peak systolic velocity	−6.15 ± 12.41	1.79 ± 12.69	−5.34 ± 14.83	−2.55 ± 13.48	0.15
S/D ratio	−0.13 ± 0.46	−0.25 ± 0.6	−0.01 ± 0.52	−0.15 ± 0.54	0.94
Pulsatility index	−0.05 ± 0.2	−0.072 ± 0.2	−0.03 ± 0.15	−0.05 ± 0.18	0.91
Resistance index	−0.01 ± 0.01	0.01 ± 0.16	−0.11 ± 0.14	−0.03 ± 0.07	0.94
Time-averaged maximum velocity	−3.1 ± 8.7	3.23 ± 19.15	−5.57 ± 20.07	−1.17 ± 17	0.16
End diastolic velocity	−1.54 ± 5.56	1.79 ± 5.99	−1.76 ± 6.62	−0.18 ± 6.16	0.17
Ductus Venosus					
S/a ratio	−2.01 ± 4.24	0.27 ± 2.43	−1.83 ± 11.08	−0.99 ± 6.43	0.97
a/S ratio	19.07 ± 13.25	16.40 ± 11.73	13.76 ± 11.83	−0.15 ± 0.54	0.77
Pulsatility index	−0.21 ± 0.63	−0.10 ± 0.9	0.31 ± 0.77	−0.026 ± 0.8	0.65
Peak velocity index for veins	0.93 ± 4.55	0.11 ± 1.17	−8.37 ± 32.54	−2.11 ± 17.41	0.94
Heart rate	−7 ± 2.7	7.24 ± 15.37	3.67 ± 18.22	2.09 ± 20.01	0.62
Peak velocity index for veins	0.93 ± 4.55	−0.11 ± 1.17	−8.375 ± 32.54	−2.11 ± 17.41	0.62

## Data Availability

Data may be obtained upon reasonable request to the corresponding author.

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
