# Peer review of "The Effect of External Cephalic Version on Fetal Circulation: A Prospective Cohort Study"

_children, 2023, doi:10.3390/children10020354_

Round 1
Reviewer 1 Report
Described medical problem is really important and many Ob/Gyns as truly interested in safety and outcomes of this procedure. Presented paper has many flaws which make this paper not very convincing in manner of Doppler examination in uncomplicated pregnancy... I have enlisted my observations. Answering those questions will improve quality of revised paper. In general I would deeply reformulate the paper taking into account my suggestions.
1. Ref. 2. You cite not up-to-date ACOG recommendation. Number 221 from 2020. It must be changed.
2. Percentages in lines 42-43 have to attached to particular Ref.
3. Ref. 1. You cite paper from 2015 and point it to 2010 Cochrane review... Moreover, in lines 44-46 you mention data which are not true... Has to changed.
4. In line 61-63 you mention about parameters analysed in the Doppler exam, I would expect to have enlisted indications for this type of examination. Moreover it would be useful to explain what is the rationale of performance such of examination in ECV -> Pathogenesis of complications as a result of ECV...?!
5. The paper by the Lau et al. should be the most important reference in your paper -> for comparison of the results as well for discussion. There are many differences between your and Lau's paper. Is why they performed 10 min. before and 5 m. after and you 1h before and 2h after? What is the rationale? Why you didn't perform such test 24-48h after?? This would improve quality and answer the question if the results are acute or persistent?...
6. Data in methodology are missing: gate, timing of exam etc.
7. There are no criteria of inclusion and exclusion.
8. Prior the first attempt there was no b-bloker?
9. In line 88 "until satisfactory" - if it wasn't you performed 2,3 attempt or qualified for elective cs?
10. Statistics should be expanded. Why only Wilcoxon test was performed? Also in line with what I mentioned; paper of Lau et al. Level of significance was p<0.05?
11. Table 1. is predominantly part of group characteristics, this are materials, not results...
12. It is confusing. Succes rate was 75%,but you performed 24 cs? So who are those 10 pregnant ? Succesful ECV with emergency CS? It is not explained....
13. Not obvious in table 2. Only succesful cases are presented? Routine vs. complicated (with 1 or 2 attempts)? Also those succesful were qualified for CS for emergency reasons?
14. Definition of difficulty of ECV is not presented, should be explained. What does it mean complicated, 2-3 attempts? etc.
15. In table 15 there are two RI in UA. But it is confusing which one is ok?
16. In line 154 you mentioned about complications. 1/56 abruptions in rather not "only" - approx. 2% so 2 x more frequently than usual... 1/56 inversion -> very rare 1:2000-1:50000....
17. Limitations are important part of every work. You mentioned about quantity of the group, that's true. As you wrote that this is not routinely performed study -> this is not study limitation. This is the main aim of the study... Limitation or rather disadvantage of your paper is that you didn't perform analysis after 24-48h after ECV... This is really weak point, as we cannot judge if this is short or long term result...
18. References should be refreshed. Deeply.
Reassessment after substantial revision.
Author Response
Reviewer 1 ECV Described medical problem is really important and many Ob/Gyns as truly interested in safety and outcomes of this procedure. Presented paper has many flaws which make this paper not very convincing in manner of Doppler examination in uncomplicated pregnancy... I have enlisted my observations. Answering those questions will improve quality of revised paper. In general I would deeply reformulate the paper taking into account my suggestions. Response: We thank the reviewer for his extensive and comprehensive review. The reviewer's suggestions improved the paper substantially. 1. Ref. 2. You cite not up-to-date ACOG recommendation. Number 221 from 2020. It must be changed. Response: The reference was updated, line 41 2. Percentages in lines 42-43 have to attached to particular Ref. Response: The reference was added (lines 44-46). 3. Ref. 1. You cite paper from 2015 and point it to 2010 Cochrane review... Moreover, in lines 44-46 you mention data which are not true... Has to changed. Response: Line 44, this was a typing error, the relevant Cochrane review is the one cited (ref 1). We thank the reviewer for noting this and revised and corrected it (lines 46-48). 4. In line 61-63 you mention about parameters analyzed in the Doppler exam, I would expect to have enlisted indications for this type of examination. Moreover it would be useful to explain what is the rationale of performance such of examination in ECV -> Pathogenesis of complications as a result of ECV...?! Response: We agree with the reviewer and added the indications for fetal circulation doppler examination, along with the rationale for performing a doppler examination after ECV (lines 55-60 and lines 75-80). 5. The paper by the Lau et al. should be the most important reference in your paper -> for comparison of the results as well for discussion. There are many differences between your and Lau's paper. Is why they performed 10 min. before and 5 m. after and you 1h before and 2h after? What is the rationale? Why you didn't perform such test 24-48h after?? This would improve quality and answer the question if the results are acute or persistent?... Response: Thank you for this interesting comment. We aimed to evaluate the early fetal Doppler response to ECV and not the immediate response, as published by Lau et al. The early response, which was defined as 2 hours difference has a stronger effect on fetal circulation compared to immediate change that might disappear immediately. An additional examination later, would reflect persistent flow changes that were not part of the scope of this study. We agree that an additional examination later, would have expand our understanding of the maximum duration of the changes, but that was not the aim of this study. However, this is one of the main limitations of our study, which we state in lines 262-265. 6. Data in methodology are missing: gate, timing of exam etc. Response: Information about the gate and timing were added (lines 122-126). 7. There are no criteria of inclusion and exclusion. Response: Inclusion and exclusion criteria were added (lines 93-98). 8. Prior the first attempt there was no b-bloker? Response: No, the b-blocker was not given routinely, but only after a preliminary attempt failed and a clinical impression of the performing physician that tocolysis might assist the procedure. This information was added (lines 109-110). 9. In line 88 "until satisfactory" - if it wasn't you performed 2,3 attempt or qualified for elective cs? Response: Our protocol required continuous monitoring during ECV. As long as the monitoring was satisfactory, we performed 2-3 attempts, up to the point where a clinical impression determined failure. In case of severe, acute fetal distress we would perform urgent cs. Happily, none of the patients in our study met this indication (line 150). 10. Statistics should be expanded. Why only Wilcoxon test was performed? Also in line with what I mentioned; paper of Lau et al. Level of significance was pReviewer 2 Report
The idea is interesting but the manuscript needs major revisions:
- erase the table and anything related to the difficulty of the procedure
- redo the analysis based on this: which fetuses had a significant changed in Doppler values? Compared this babies with the babies who did not have significant changes, regarding the pregnancy outcome
- rewrite the entire manuscript basing it ONLY on these 2 purposes
Author Response
Reviewer 2 The idea is interesting but the manuscript needs major revisions: - erase the table and anything related to the difficulty of the procedure Thank you for your remark. One of the main questions that led to this study was whether the difficulty in performing the procedure might influence the response of the fetus, as expressed by the Doppler indices. This issue was not discussed in the literature and we believe that this new information will be helpful to clinicians performing ECV. - redo the analysis based on this: which fetuses had a significant changed in Doppler values? Compared this babies with the babies who did not have significant changes, regarding the pregnancy outcome This study aimed to describe changes in Doppler indices in the early period following the ECV and to determine whether the difficulty performing the procedure influenced the Doppler flow results. Hence, we focused on the Doppler measurements in the cases studied. We would be happy and interested to adopt your suggestion and analyze the results in a future study. - rewrite the entire manuscript basing it ONLY on these 2 purposes We have made some corrections and revisions according to your remarks. Rewriting the entire manuscript would miss the purpose and the results of this study. We will definitely adopt your remarks when conducting further studies regarding ECV.Reviewer 3 Report
10-01-2023.
The article describes effects of external cephalic version on fetal circulation.
The article is clear and well written.
I ahev a few typo’s and questions.
About the typo’s:
Line 33: n = s breech presentation is etc.
Line 136. It is better to put after this result that it is table 2.
Line 181: ECF should be ECV?
Line 223: The ECVB was performed.
My questions are:
In the introduction you say that ECV is becoming more common. Line 72. On what is that opinion based?
Line 145: there was a significant correlation the level of difficulty of the ECV procedure and between the need for manual lysis of the placenta at birth. Was there more infection?? , since that is often the reason for manual lysis of the placenta.
Otherwise congratulations with this nice article.
Author Response
Reviewer 3 The article describes effects of external cephalic version on fetal circulation. The article is clear and well written. I have a few typo’s and questions. About the typo’s: Line 33: n = s breech presentation is etc. Line 136. It is better to put after this result that it is table 2. Line 181: ECF should be ECV? Line 223: The ECVB was performed. Answer: Many Thanks for these remarks, we corrected them as necessary. My questions are: In the introduction you say that ECV is becoming more common. Line 72. On what is that opinion based? Answer: We followed reference number 2 who stated this observation. Line 145: there was a significant correlation the level of difficulty of the ECV procedure and between the need for manual lysis of the placenta at birth. Was there more infection?? , since that is often the reason for manual lysis of the placenta. Answer: Thank you for this important remark, unfortunately we did not check placental infection as we were not aware of the association of ECV and later infection. Otherwise congratulations with this nice article. Answer: Thank you for this encouraging note.Round 2
Reviewer 1 Report
Thank you for the revision. I see great improvement of the presented manuscript. I still have some concerns about soundness and impact of the results of provided study. It would be really helpful if you would compare results short and long term impact. I would add this sentence to the manuscript that it may be useful, crucial to forward, to provide subsequent studies in this field. It might be a good idea to add this info in the title of the paper.